# miR-182-5p Regulates Nogo-A Expression and Promotes Neurite Outgrowth of Hippocampal Neurons In Vitro

**DOI:** 10.3390/ph15050529

**Published:** 2022-04-25

**Authors:** Altea Soto, Manuel Nieto-Díaz, David Reigada, María Asunción Barreda-Manso, Teresa Muñoz-Galdeano, Rodrigo M. Maza

**Affiliations:** Molecular Neuroprotection Group, Research Unit, National Hospital for Paraplegics (SESCAM), 45071 Toledo, Spain; alteas@sescam.jccm.es (A.S.); dreigada@sescam.jccm.es (D.R.); mbarreda@sescam.jccm.es (M.A.B.-M.); tmunozd@sescam.jccm.es (T.M.-G.)

**Keywords:** miR-182-5p, Nogo-A, neurite outgrowth, axonal regeneration, spinal cord injury, neurodegenerative diseases

## Abstract

Nogo-A protein is a key myelin-associated inhibitor of axonal growth, regeneration, and plasticity in the central nervous system (CNS). Regulation of the Nogo-A/NgR1 pathway facilitates functional recovery and neural repair after spinal cord trauma and ischemic stroke. MicroRNAs are described as effective tools for the regulation of important processes in the CNS, such as neuronal differentiation, neuritogenesis, and plasticity. Our results show that miR-182-5p mimic specifically downregulates the expression of the luciferase reporter gene fused to the mouse Nogo-A 3′UTR, and Nogo-A protein expression in Neuro-2a and C6 cells. Finally, we observed that when rat primary hippocampal neurons are co-cultured with C6 cells transfected with miR-182-5p mimic, there is a promotion of the outgrowth of neuronal neurites in length. From all these data, we suggest that miR-182-5p may be a potential therapeutic tool for the promotion of axonal regeneration in different diseases of the CNS.

## 1. Introduction

During postnatal development, central nervous system (CNS) neurons lose their ability to regenerate, in part due to the presence of myelin-derived inhibitors of axonal outgrowth and neuroregeneration, such as MAG, OMGp, and Nogo-A [1]. The *RTN4* gene (Nogo) belongs to the family of reticulon-encoding genes that produces three major protein variants, RTN4A (Nogo-A), RTN4B (Nogo-B), and RTN4C (Nogo-C), by alternative splicing, promoter usage, and alternative polyadenylation, respectively [2]. They all share a common 188-amino-acid C-terminal membrane-spanning domain known as a reticulon homology domain (RHD domain) consisting of two hydrophobic regions flanking a hydrophilic loop (Nogo-66), which is followed by a short C-terminal tail. Nogo-A is the largest of the Nogo isoforms, with two known neurite growth inhibitory domains, including amino-Nogo (Nogo-A-Δ20) at the N terminus and the extracellular loop Nogo-66 [3].

Nogo function depends on the tissular expression of each isoform [4]. Nogo-A has been studied extensively in the CNS and is responsible for the inhibition of neurite outgrowth, acting as a myelin-associated inhibitor of axon regeneration. Nogo-A is mostly expressed in spinal motor, DRG, sympathetic, hippocampal, and Purkinje neurons, as well as oligodendrocytes. In the spinal cord, Nogo-A is mainly expressed in grey matter, especially by large motor neurons of the ventral horns [5,6,7]. The functions carried out by Nogo-A in neurons and oligodendrocytes appear to be quite different. Whereas neuronal Nogo-A seems to act as a direct local restrictor of synaptic and dendritic plasticity, oligodendrocytic Nogo-A may act as an inhibitor of axonal growth by binding to its receptor (Nogo receptor, NgR1) located on neurons. This binding transduces the inhibitory signal to the cell interior via transmembrane co-receptors LINGO-1 and p75NTR or TROY, by which the small GTPase RhoA is activated [1,3,8,9,10]. Multiple studies have demonstrated the efficacy of targeting the Nogo-A/NgR1 pathway for functional recovery and neural repair after spinal cord trauma, ischemic stroke, optic nerve injury, and models of multiple sclerosis [11].

MicroRNAs (miRNAs) are an abundant class of small non-coding RNAs that operate as epigenetic modulators of gene expression in physiology but also in pathophysiological processes. They are involved in post-transcriptional gene silencing by base pairing to their target mRNAs of protein-coding genes, resulting in reduced translation of the protein by mRNA repression or degradation [12,13]. The flexibility and efficiency of miRNA function provide both spatial and temporal gene regulatory capacities that are essential for establishing neural networks. The expression of miRNAs is ubiquitous in neural tissues, and many regulate neuronal differentiation, neuritogenesis, excitation, synaptogenesis, and plasticity [13,14,15]. There is a close relationship between miRNAs and intrinsic determinants of axonal regeneration. Several miRNAs have been proven to regulate the expression levels of targets involved in neurite outgrowth and axonal regeneration after CNS injury. For example, miR-431 is nerve-injury-induced miRNA that stimulates regenerative axon growth by silencing Kremen1, an antagonist of Wnt/beta-catenin signaling [16] or miR-133b, that has also been shown to promote neurite outgrowth in primary cortical neurons and PC12 cells by targeting RhoA [17]. Other examples are miR-124, a well-conserved brain-specific miRNA that promotes neurite outgrowth of M17 cells by targeting ROCK1 GTPase [18], or miR-222, which regulates neurite outgrowth from DRG neurons by targeting PTEN [19].

However, a functionally validated miRNA that regulates the expression of Nogo-A has not been yet described. Although the reticulon (RTN) family isoforms mature through splicing and alternative polyadenylation processes, the *RTN4* gene shares the highly conserved carboxy-terminal reticulon domain and 3′UTR. We thus seek functional validated miRNAs capable of inhibiting the activity of the Nogo family members, and only miR-182-5p was found to directly target Nogo-C 3′UTR and decrease Nogo-C protein levels in cardiomyocytes cells [20]. miR-182-5p is a member of the miR-183 family located on chromosome 7q31-34 and is described as an oncogenic miRNA due to its capacity to enhance cancer cell proliferation, survival, tumorigenesis, and drug resistance [21,22]. Although miR-182-5p roles are well known in cancer, little is known about its function in the CNS under normal and pathophysiological conditions. Wang and colleagues demonstrated that miR-182-5p promotes axonal growth and regulates neurite outgrowth via the PTEN/AKT pathway in cortical neurons [23]. Moreover, miR-182-5p is the most abundant miRNA in retinal ganglion cell axons, where it regulates Slit2-mediated axon guidance in vitro and in vivo [24]. Furthermore, Yu and colleagues showed that miR-182-5p inhibits Schwann cell proliferation and migration by targeting FGF9 and NTM in sciatic nerve injury [25].

Because all Nogo protein variants share the conserved carboxy-terminal reticulon domain and 3′UTR, we hypothesize that miR-182-5p could also regulate Nogo-A expression. In the present study, we perform a bioinformatic and validation characterization of the miR-182-5p site in the Nogo-A 3′UTR to demonstrate for the first time that miR-182-5p downregulates Nogo-A protein expression in Neuro-2a and C6 cells and promotes neurite outgrowth of rat primary hippocampal neurons in vitro.

## 2. Results

### 2.1. miR-182-5p Is Predicted to Regulate Mouse, Rat, and Human Nogo-A 3′UTRs

A bioinformatics-based prediction of the potential targets of miR-182-5p in mouse mRNAs was performed. Because the various available programs can yield rather different predictions, we combined miRmap, miRanda 3.3a, TargetScan 8.0, and miRWalk 3.0 programs to search for mouse miR-182-5p gene targets (Figure 1A). A total of 191 common genes were identified by the four prediction programs, including Nogo (*RTN4* gene).

According to the employed prediction programs, the Nogo-A 3′UTR of all species has one binding site for miR-182-5p. Alignment of this putative site in rat, mouse, and human sequences demonstrates the evolutionary conservation of the Nogo-A site among mammalian species (Figure 1B). Because the three Nogo protein variants share the same 3′UTR region, we observe that this site matches with the already validated miR-182-5p site in the Nogo-C 3′UTR [20]. Analyses of target site accessibility of the mRNA secondary structure by STarMir tool further support miR-182-5p targeting on the human Nogo-A 3′UTR. The logistic probability (LogitProb) of the Nogo-A 3′UTR site being a miR-182-5p binding site is 0.635 (Figure 1C). In general, a LogitProb of 0.5 indicates a fairly good chance of miRNA binding [26]. The hybrid diagram of the seed site (Figure 1C) shows that miR-182-5p has a canonical site at the human Nogo-A 3′UTR, with a hybridization energy (ΔGhybrid) of −27.10 kcal/mol; therefore, the interaction between miR-182-5p and its seed region in the human Nogo-A 3′UTR is thermodynamically stable.

Taken together, the bioinformatics approach supports the potential of miR-182-5p as a site in the sequence of Nogo-A 3′UTR in the three species studied; therefore, miR-182-5p can play a biologically relevant role in regulating their expression.

### 2.2. Nogo-A and miR-182-5p Expression in Neural Cell Lines

We chose Neuro-2a and C6 cell lines to study Nogo-A regulation by miR-182-5p mimic in vitro due to their Nogo-A protein expression levels (Figure 2A) and endogenous expression of miR-182-5p (Figure 2B). Conversely, SH-SY5Y cell line was discarded because Nogo-A protein is undetected in its extracts.

### 2.3. miR-182-5p Targets the Mouse Nogo-A 3′UTR and Downregulates Its Protein Expression

The luciferase activity of the pmiRGLO plasmid in the presence of miR-182-5p mimic was evaluated to rule out any effect of the miRNA on plasmid expression, and no significant effects were detected (Figure 3A) (109.1 ± 21.71% vs. 100% empty pmiRGLO without mimic co-transfection; two-tailed paired *t*-test, T_4_ = 0.4203, n.s. *p* = 0.6958).

In addition, Negative control mimic co-transfection does not affect the firefly/Renilla emission ratio of pmiRGLO-3′UTR-Nogo-A-wt (115.24 ± 11.17% vs. 100% pmiRGLO-3′UTR-Nogo-A-wt without mimic co-transfection; two-tailed paired *t*-test, T_4_ = 1.365, n.s. *p* = 0.2441) (Figure 3A). However, co-transfection with miR-182-5p mimic causes a significant reduction in the emission ratio of 43.33% compared to co-transfection with the negative control mimic (71.87 ± 4.01% vs. 115.24 ± 11.17% negative control mimic; two-tailed paired *t*-test, T_4_= 4.789, *p* = 0.0087).

Then, the effects of both mimics on pmiRGLO-3′UTR-Nogo-A-mut plasmid were evaluated using a mutation in the predicted miR-182-5p binding site to validate that the effect of miR-182-5p reducing the emission ratio is due specifically to the interaction with its binding site. Co-transfection with the negative control mimic (117.8 ± 14.01% vs. 100% pmiRGLO-3′UTR-Nogo-A-mut; two-tailed paired *t*-test, T_4_ = 1.273, n.s. *p* = 0.2719) or miR-182-5p mimic (116.1 ± 14.52% vs. 100% pmiRGLO-3′UTR-Nogo-A-mut; two-tailed paired *t*-test, T_4_ = 1.107, n.s. *p* = 0.3304) had no effect on the luciferase emission ratio (Figure 3A).

Finally, to evaluate the modulation of Nogo-A protein expression levels by miR-182-5p, Neuro-2a and C6 cells were transfected with either miR-182-5p or negative control mimics for 24 h, and protein expression levels were detected by immunoblot assay. Transfection with miR-182-5p mimic significantly downregulated the endogenous Nogo-A protein levels compared to negative control mimic transfection in both the Neuro-2a cell line (70.98 ± 3.98% vs. 100% Negative control mimic; two-tailed paired *t*-test, T_2_ = 7.291, *p* = 0.0183) and the C6 cell line (63.51 ± 7.67% vs. 100% Negative control mimic; two-tailed paired *t*-test, T_2_ = 4.755, *p* = 0.0415) (Figure 3B).

### 2.4. Nogo-A Downregulation by miR-182-5p Mimic Promotes Neurite Outgrowth of Rat Primary Hippocampal Neurons

To determine the biological effects of the downregulation of Nogo-A by miR-182-5p transfection on neurite outgrowth, we performed functional analyses based on the co-culture of rat primary hippocampal neurons with C6 cells transfected with either miR-182-5p or negative control mimics. Transfection of C6 cells with miR-182-5p mimic significantly increased neurite length of primary hippocampal neurons by ~26 µm per neuron (miR-182-5p co-culture) in comparison with the negative-control-transfected C6 cells (negative control co-culture) (183.0 ± 20.32 µm vs. 157.7 ± 20.59 µm negative control co-culture; two-tailed paired *t*-test, T_2_ = 17.65, *p* = 0.0032) (Figure 4A,B). Moreover, transfection of C6 cells with both negative control and miR-182-5p mimics increased neurite length in comparison with the non-transfected C6 cells (control co-culture). Negative control mimic co-culture induced a significant increase of ~17 µm per neuron (157.7 ± 20.59 µm vs. 140.1 ± 17.58 µm control co-culture; two-tailed paired *t*-test, T_2_ = 5.041, *p* = 0.0372), and miR-182-5p mimic co-culture produced an increase of ~42 µm per neuron (183.0 ± 20.32 µm vs. 140.1 ± 17.58 µm control co-culture; two-tailed paired *t*-test, T_2_ = 15.46, *p* = 0.0042).

Differences in cell density were observed in the transfected cultures. Transfection of both mimics significantly reduced the C6 cell density in the co-cultures compared to non-transfected C6 cell co-culture (control co-culture) (two-way ANOVA, F_2,12_ = 25.52, *p* < 0.0001; Tukey’s multiple comparison post hoc test, *p* < 0.0001), although hippocampal neuronal densities were not significantly changed (Figure 4C).

## 3. Discussion

Amongst many factors, one of the major inhibitory signals of the CNS environment to regrowth is the myelin-associated Nogo pathway, which plays an important role in regeneration [27,28,29]. Oligodendrocytic Nogo-A binds to its neuronal receptor (NgR1) and co-receptors (LINGO-1 and p75NTR or TROY), inhibiting neurite outgrowth of the neurons [8,9,10]. In the present study, we describe and validate, for the first time, Nogo-A post-transcriptional regulation by a miRNA in two murine neural cell lines, Neuro-2a and C6, which have been extensively used to study neuronal differentiation and axonal growth [30,31]. We demonstrate that miR-182-5p downregulates Nogo-A expression, promoting neurite outgrowth of primary hippocampal neurons in vitro.

The involvement of Nogo-A in neurodegeneration has been described in diverse CNS diseases, such as ocular diseases, multiple sclerosis, Alzheimer’s disease, and amyotrophic lateral sclerosis, as well as spinal cord injury (SCI) and traumatic brain injury. However, the role of Nogo-A is not restricted to the CNS; Nogo-A also inhibits the spread and migration of non-neuronal cell, such as fibroblasts and vascular endothelial cells [32,33].

The dysregulation of Nogo-A following CNS injury, in particular SCI, is in line with the expression changes of numerous genes that play vital roles in the pathogenesis of secondary CNS damage or axonal regeneration [34,35]. Most of these genes are regulated by the post-transcriptional regulator miRNAs [36], which showed an altered expression following injury [37,38,39,40,41]. Both in vitro and in vivo evidence has demonstrated that these miRNAs participate in crosstalk with key genes involving processes of neuronal plasticity, neuronal degeneration, axonal regeneration, remyelination, and glial scar formation after SCI through translational repression or mRNA degradation [42,43,44,45]. Thus, miR-133b, miR-135a-5p, and miR-29a regulate neurite outgrowth and axon regeneration by targeting RhoA, ROCK1/2, and PTEN genes, respectively. Moreover, the overexpression of these miRNAs has been shown to contribute to spinal cord regeneration and functional recovery in murine SCI models [17,46,47,48]. Similarly, miR-182-5p has been found to be involved in secondary damage of CNS processes and neuronal regeneration. According to miRNATissueAtlas2 (https://ccb-web.cs.uni-saarland.de/tissueatlas2, accessed on 13 January 2022; [49]), miR-182-5p is mainly expressed in blood vessels, the epididymis, and the CNS, especially in the spinal cord.

Previous analysis from our laboratory [38] and others [37,50,51] revealed miR-182-5p as one of the most downregulated miRNAs after injury, in agreement with recently described time-course miR-182-5p expression results in SCI [40]. In this study, the highest downregulation point of miR-182-5p expression was observed at 7 days post injury, with expression recovery at 28 days, which interestingly parallels both Nogo-A protein and mRNA expressions, which rapidly rose to a peak after 7 days and then gradually declined again after 14 days [52]. However, to dare, a validated miRNA targeting Nogo-A has not been reported. In accordance with our studies, the modulatory role of miR-182-5p on Nogo-A, which was validated by our luciferase and immunoblot results (Figure 3), could explain this dynamic of expression changes following injury. The validity of the miR-182-5p and Nogo-A interaction is supported by the miR-182-5p regulation of another member of the *RTN4* family, namely Nogo-C, which shares 3′UTR with Nogo-A.

Overexpression of miR-182-5p targets the Nogo-C 3′UTR and decreases its protein levels, protecting cardiomyocytes from apoptosis and preserving cardiac function after myocardial infarction [20]. However, single genes may produce a variety of mRNA isoforms by mRNA modification, such as alternative polyadenylation or splicing, which could alter the selective recruitment of miRNAs to the 3′UTR [53,54]. It has been observed that nearly all genes have multiple alternative polyadenylation signals located at different positions in the 3′UTR [55]. Thus, we approached the validation of the miR-182-5p as a regulator of Nogo-A, as in both mouse and human *RTN4* gene, it has been described at more than one putative poly(A) signal site downstream of the stop codon, and the miR-182-5p site is located between these polyadenylation signals. Our bioinformatics analyses confirmed that miR-182-5p binding site on Nogo-A 3′UTR is conserved across different mammalian species, including humans. This miR-182-5p binding site in human Nogo-A 3′UTR has been confirmed by the STarMir CLIP-based tool, supporting the possibility of Nogo-A regulation by miR-182-5p in human cells. Our experimental data concerning reporter gene regulation and Nogo-A endogenous expression levels after overexpression of miR-182-5p in cell cultures validate this microRNA response element in the Nogo-A 3′UTR.

Although the role of miR-182-5p as a regulator of neurite outgrowth has been described in cortical and midbrain neurons through activation of the PTEN/AKT pathway [23,56], our results could provide a broader implication with regards to axonal regeneration. Our functional assays (Figure 4) showed that the downregulation of Nogo-A due to miR-182-5p overexpression in neural cells eased the neurite outgrowth of primary hippocampal neurons. However, a better understanding of miR-182-5p regulation on Nogo-A, including exploring non-canonical mechanisms (e.g., paracrine regulation by miR-182-5p expressing cells), is needed to establish its precise role following CNS injury. Analyses would greatly benefit from gain- and loss-of-function assays employing stable and conditional cell lines.

Neutralizing Nogo-A by function-blocking antibodies or genetic knockout (KO) has been shown to improve axonal sprouting and regeneration in the injured spinal cord and brain [11,57,58], and the clinical potential of anti-Nogo-A antibodies for managing SCI is currently being investigated in two clinical trials (ClinicalTrials.gov accessed on 25 January 2022|Identifiers: NCT03935321 and NCT03989440). Therefore, Nogo-A downregulation by overexpression of miR-182-5p could be a potential treatment for different diseases and conditions that implicate axonal degeneration.

## 4. Materials and Methods

### 4.1. Bioinformatics and Data Mining

To identify miR-182-5p response elements in mouse messenger RNAs (mRNA), an in silico screening approach was used. This approach combines computational tools that employ existing databases and prediction algorithms and data mining for gene expression data analysis. The four prediction tools used were miRMap (https://mirmap.ezlab.org/, last accession on 2 November 2021), miRanda 3.3a (http://www.microrna.org, accessed on 2 November 2021), TargetScan 8.0 (http://www.targetscan.org, accessed on 3 November 2021), and miRWalk 3.0 (http://mirwalk.umm.uni-heidelberg.de/, accessed on 3 November 2021). 

miR-182-5p binding site accessibility on the human Nogo-A 3′UTR (3′UTR-Nogo-A) was analyzed using the STarMir tool [26], an implementation of logistic prediction models developed with miRNA binding data from crosslinking immunoprecipitation (CLIP) studies. In the STarMir web (https://sfold.wadsworth.org/cgi-bin/starmirtest2.pl, accessed on 5 November 2021), we input hsa-miR-182-5p into the option of “microRNA sequence(s), microRNA ID(s)”. The human 3′UTR-Nogo-A sequence was input into the option of “single target sequence, manual sequence entry”. With the choice of “V-CLIP based model (human)”, “Human (homo sapiens)”, and “3′UTR”, a set of parameters (described in http://sfold.wadsworth.org/data/STarMir_manual.pdf accessed on 5 November 2021) was displayed in the output window for further analysis.

### 4.2. Cell Lines and Cultures

Neuro-2a mouse neuroblastoma cells (cat#: CCL-131, ATCC; RRID#CVCL_0470) and C6 rat brain glioma cells (cat#: CCL-107, ATCC; RRID##CVCL_0194) were cultured in Dulbecco’s modified Eagle medium (DMEM; Gibco) supplemented with 10% fetal bovine serum (FBS; Gibco), 1% penicillin/streptomycin (Gibco), and 1% glutaMAX (Gibco). SH-SY5Y human neuroblastoma cells (cat#: CRL-2266, ATCC; RRID#CVCL_0019) were grown in a 1:1 combination of minimum essential medium (MEM; Gibco) and Ham’s F-12 nutrient mixture (Gibco) supplemented with 10% FBS, 1% penicillin/streptomycin, 1 mM sodium pyruvate (Gibco), and non-essential amino acids (Gibco). Cells were cultured at 37 °C in a humidified incubator containing 5% CO_2_.

Primary hippocampal neurons were obtained from 17–18-day-old (E17-18) Wistar rat embryos. Briefly, after dissection, hippocampi were subjected to enzymatic digestion in Hanks′ balanced salt solution (HBSS) medium without calcium or magnesium (Hyclone, GE Healthcare) supplemented with trypsin (1×; Thermo Fisher Scientific) and DNAse (20 mg/mL; Roche) for 15 min at 37 °C. Trypsin and DNAse were washed out with HBSS, with calcium and magnesium (Hyclone) and mechanically disrupted by passing the tissue sample several times through a glass pipette in neurobasal medium (Gibco) supplemented with 2% B-27 supplement (Gibco), 1% GlutaMAX (Gibco), and 1% penicillin/streptomycin. Neurons were cultured on C6 cells at 37 °C in a humidified incubator containing 5% CO_2_.

### 4.3. RNA Extraction and Quantitative Real-Time PCR (RT-qPCR)

Total RNA was isolated from Neuro-2a, C6, and SH-SY5Y cells with an miRNeasy Kit (Qiagen) and was analyzed with a NanoDrop ND-1000 spectrophotometer (Thermo Fisher Scientific) to determine its concentration and purity (260/280 and 260/230 ratios). To determine cellular miR-182-5p expression, 10 ng of total RNA was reverse-transcribed and amplified using TaqMan miRNA gene expression assay (TaqMan^®^ MicroRNA assay #002284, Applied Biosystems) following the manufacturer’s protocols. The U6 small nuclear RNA served as an internal control (TaqMan^®^ MicroRNA assay #001973, Applied Biosystems). The abundance of the miRNA was measured in a thermocycler ABI Prism 7900 fast (Applied Biosystems) applying 40 cycles of a two-step protocol: 15 s at 95 °C plus 1 min at 60 °C. The ΔCt value was defined as the difference between the cycle threshold of amplification kinetics (Ct) of the target miRNA and its U6 loading control [59].

### 4.4. Vector Construction

The sequence of the mouse Nogo-A 3′UTR (NM_194054.3) (3′UTR-Nogo-A) containing the predicted binding site for mmu-miR-182-5p (nt 4272–4279) wild type (wt) was obtained by amplification by PCR from mouse brain total RNA preparation (Table 1). The 3′UTR-Nogo-A-wt product was subcloned into both the pGEM-T-easy plasmid (Promega) and the pBKS plasmid (pBluescript, Stratagene). The 3′UTR-Nogo-A-wt sequence was validated by DNA sequencing (T7p and SP6) and inserted into the pmiRGLO dual-luciferase miRNA target expression vector (Promega, http://www.addgene.org/vector-database/8236/ accessed on 16 November 2021) between the SacI and XbaI restriction sites (pmiRGLO-3′UTR-Nogo-A-wt) using the FastDigest restriction enzymes (Thermo Fisher Scientific). The miR-182-5p mutant site of 3′UTR-Nogo-A (3′UTR-Nogo-A-mut) was obtained by PCR using the 3′UTR-Nogo-A-wt subcloned into pBKS plasmid as template, specific primers (Table 1), and PfuI polymerase (Thermo Fisher Scientific), followed by DpnI endonuclease restriction digest (FastDigest, Thermo Fisher Scientific) and a transformation in *E. coli* supercompetent cells (Thermo Fisher Scientific). The 3′UTR-Nogo-A-mut fragment was inserted into the pmiRGLO plasmid between the SacI and SalI restriction sites (pmiRGLO-3′UTR-Nogo-A-mut). Finally, we confirmed the sequence of both pmiRGLO 3′UTR-Nogo-A constructs by DNA sequencing using a specific forward 3′ end luciferase primer.

### 4.5. Dual-Luciferase Reporter Assay

To validate the targeting of miR-182-5p mimic on mouse 3′UTR-Nogo-A, Neuro-2a cells were cultured to 70% confluence in white 96-well plates. Then, cells were co-transfected using DharmaFECT Duo transfection reagent (DharmaconTM) with either (i) 50 nM of miR-182-5p mimic (miRIDIAN; cat#: C-320575-01-0005, DharmaconTM) or 50 nM cel-miR-67 negative control mimic (Negative control mimic) (miRIDIAN microRNA mimic negative control#1; cat#: CN-001000-01; DharmaconTM) and (ii) 200 ng/well of pmiRGLO-3′UTR-Nogo-A-wt, pmiRGLO-3′UTR-Nogo-A-mut or empty pmiRGLO (without subcloned 3′UTR) as endogenous regulation control. After 24 h, the plasmid gene expression under the regulation of both Nogo-A 3′UTRs was evaluated by measuring firefly and Renilla luciferase activities using a Dual-GLO luciferase assay system (Promega) in an Infinite M200 plate reader (Tecan) according to the manufacturer’s protocol. Firefly emission data were normalized to Renilla load control levels and expressed as the firefly/Renilla ratio. All experiments were performed in triplicate in at least five independent experiments.

### 4.6. Immunoblot Assay

To evaluate the endogenous expression levels of Nogo-A protein in different neural cell lines, Neuro-2a, C6, and SH-SY5Y cells were cultured to 70% confluence in 24-well plates. To evaluate the regulation of miR-182-5p mimic on Nogo-A expression, Neuro-2a and C6 cell cultures were transfected using DharmaFECT 4.0 transfection reagent (DharmaconTM) with either 50 nM of miR-182-5p or negative control mimics. After 24 h, cells were lysed with radioimmunoprecipitation assay lysis buffer (RIPA, Sigma-Aldrich) containing a complete EDTA-free protease inhibitor cocktail (Roche) for 30 min at 4 °C and cleared by centrifugation (12,000× *g*/10 min/4 °C). Protein concentration was determined by the bicinchoninic acid method (BCA protein assay kit, Thermo Fisher Scientific). An amount of 25 μg of protein was resolved by 8% dodecyl sulfate-polyacrylamide gel electrophoresis (SDS-PAGE) and then electrophoretically transferred to a 0.45 μm polyvinylidene difluoride membrane (PVDF; Immobilon, Merck Millipore). Membranes were blocked in blocking buffer (5% non-fat milk or 5% FBS diluted in TBS-T buffer (0.05% Tween-20 (Sigma-Aldrich) in tris-buffered saline (TBS)) for 30 min at 37 °C and immunoblotted with a rabbit polyclonal antibody against Nogo-A (1:100; Santa Cruz Biotechnology, cat#: sc-25660, RRID:AB_2285559) overnight at 4 °C. Mouse monoclonal antibody against α-tubulin (1:10,000; Sigma-Aldrich, cat#: T6074, RRID:AB_477582) was used as a loading control. Then, membranes were incubated with a horseradish peroxidase (HRP)-conjugated goat anti-rabbit secondary antibody (1:1000; Thermo Fisher Scientific, RRID:AB_228341) or an HRP-conjugated goat anti-mouse secondary antibody (1:1000; Thermo Fisher Scientific, RRID:AB_228307) for 1 h at room temperature (RT). Finally, protein bands were visualized using a SuperSignal West Pico chemiluminescent detection system (Pierce, Thermo Fisher Scientific, Waltham, MA, USA) and measured using ImageScanner III and ImageJ software (National Institutes of Health, Bethesda, MD, USA).

### 4.7. Neurite Outgrowth Assay

To evaluate the effect of Nogo-A regulation by miR-182-5p on neurite outgrowth of primary hippocampal neurons, different co-cultures of C6 cells (non-transfected cells, negative control mimic-transfected cells, and miR-182-5p mimic-transfected cells) and rat primary hippocampal neurons were performed. C6 cells were cultured to 70% confluence in poly-L-lysine (Sigma-Aldrich, St. Louis, MO, USA) pre-coated 24-well plates. After 24 h, cells were transfected using DharmaFECT 4.0 transfection reagent (DharmaconTM, Lafayette, CO, USA) with either miR-182-5p or negative control mimics. After 24 h, the medium was changed by supplemented neurobasal medium (described above), and 5000 primary hippocampal neurons were cultured on C6 cells for 24 h more. The co-cultures were fixed with 4% paraformaldehyde for 30 min at RT and washed with 1× phosphate-buffered saline (PBS). The cells were permeabilized and blocked with 0.2% Triton X-100 and 3% bovine serum albumin protein (BSA), respectively, in PBS for 30 min at RT. Then, cells were immunostained with a neuronally specific marker, mouse anti-β−III tubulin isoform antibody (1:500, Millipore, cat#MAB1637, RRID:AB_2210524), for 2 h at RT, followed by a goat anti-mouse Alexa Fluor 488-conjugated secondary antibody (1:500, Molecular Probes, cat# A-11029, RRID:AB_138404) for another 2 h at RT. After three washes with PBS, cell nuclei were stained with DAPI (4’,6-diamidino-2-fenilindol, 1:10,000, Merck, cat#D9542) for 5 min at RT and mounted in Lab Vision™ PermaFluor™ aqueous mounting medium (Thermo Fisher Scientific, cat#TA-030-FM). Co-cultures were imaged in an epifluorescence microscope (DM5000B, Leica Microsystem GmbH), Wetzlar, Germany) with a 40x objective and analyzed using ImageJ software.

Neurite outgrowth lengths were assessed using the method described by Rønn and colleagues [60]. Briefly, the absolute neurite length (L) per neuron was estimated by counting the number of intersections (I) between neurites and test lines of a grid superimposed on the co-culture images and the equation L = (πd/2)I, where d is the vertical distance between the test lines of the grid. The neurite length increase per neuron was calculated using the control co-culture (with non-transfected C6 cells) as reference, that is, subtracting the mean neurite length per neuron of the control co-culture from the mean neurite length per neuron of the mimic-transfected co-cultures. C6 cells and hippocampal neuronal density were analyzed by calculating the total number of each type of cell per mm^2^ in the different co-cultures (a total of nine images of 0.27 mm^2^ per condition were analyzed).

### 4.8. Statistical Analysis

Statistical significance of the transfection effects was tested using paired Student’s *t*-test or two-way ANOVA with Tukey’s multiple comparison post hoc tests. Data are expressed as mean ± SEM, as indicated in the figure legends. Statistical analyses and graphic design were conducted using GraphPad Prism version 8.0.0 (GraphPad Software). Differences were considered statistically significant when the *p*-value was below 0.05.

## 5. Conclusions

This study provides novel information about the regulatory action of miR-182-5p on Nogo-A axonal regeneration inhibition. We are the first to describe and functionally validate Nogo-A regulation by miR-182-5p, which targets Nogo-A 3′UTR, downregulates Nogo-A protein expression levels, and promotes neurite outgrowth in murine neural cell lines. Thus, miR-182-5p could be a promising therapeutic tool for diseases or conditions that implicate axonal pathology, such as SCI, brain injury, and Parkinson’s or Alzheimer’s diseases, among others.

## Figures and Tables

**Figure 1 pharmaceuticals-15-00529-f001:**
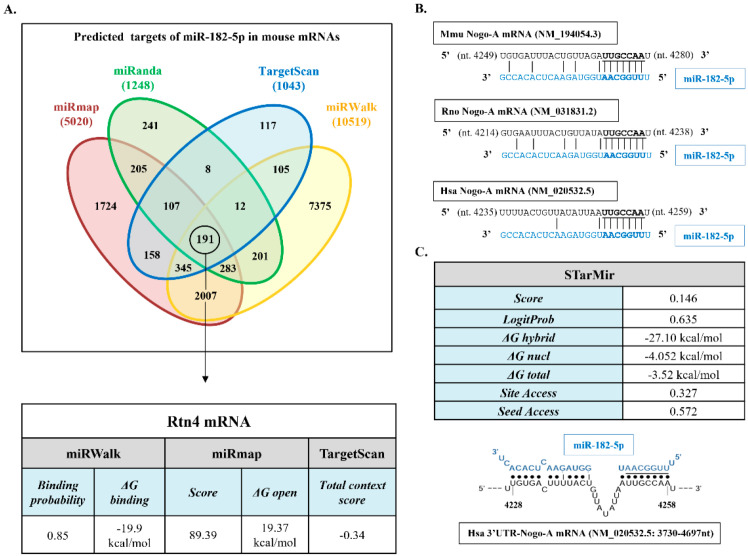
Bioinformatics analyses. (**A**) Description and main data of the process of identification of mouse *RTN4* gene (Nogo-A) as miR-182-5p-predicted target by miRmap, miRanda 3.3a, TargetScan 8.0, and miRWalk 3.0. The table shows the algorithm prediction scores, binding probability, free energy gained by transitioning from the state in which the miRNA and the target are unbound (∆G open) (kcal/mol) and the state in which the miRNA binds its target (∆G binding), according to each algorithm. (**B**) Alignment of the seed region of miR-182-5p with the Nogo-A 3′UTR in mouse (Mmu), rat (Rno), and human (Hsa) mRNAs. miR-182-5p sequence appears in blue type, and miRNA seed regions appear in bold type. (**C**) Main data of the target site accessibility and hybrid diagram seed site of miR-182-5p on human Nogo-A 3′UTR by STarMir tool. miR-182-5p sequence appears in blue type, and miRNA seed region appears underlined.

**Figure 2 pharmaceuticals-15-00529-f002:**
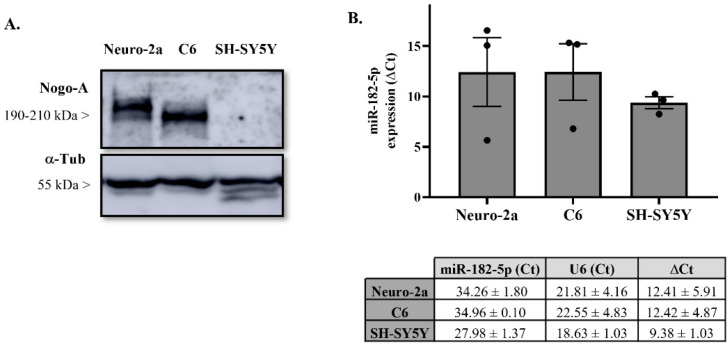
Endogenous expression levels of miR-182-5p and Nogo-A protein in different cell lines. (**A**) Representative immunoblot of Nogo-A and load-control α-tubulin protein expression in different cell line lysates of three independent experiments. (**B**) RT-qPCR showing relative miR-182-5p expression in total RNA samples isolated from different cell lines. Expression of miR-182-5p from each sample was normalized to the corresponding expression of the control gene snoRNA U6 (ΔCt = CtmiR_182_ − Ct_U6_). The bar graph and table represent the mean ± SEM data of Ct and ΔCt from three independent experiments.

**Figure 3 pharmaceuticals-15-00529-f003:**
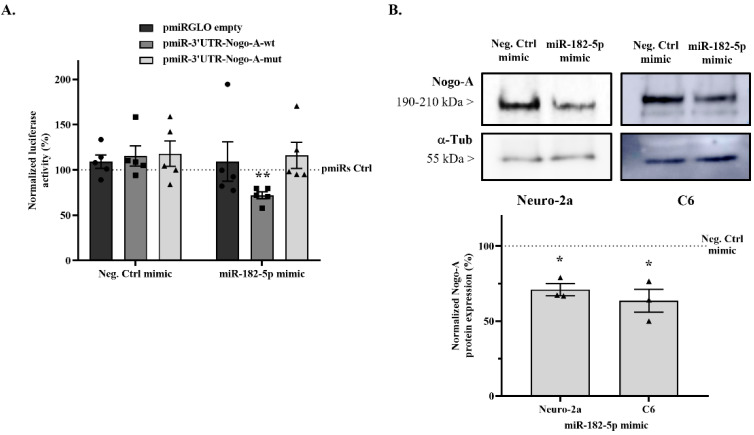
miR-182-5p mimic regulates Nogo-A expression. (**A**) Luciferase reporter activity of either miR-182-5p or negative control mimics on pmiRGLO-3′UTR-Nogo-A-wt, pmiRGLO-3′UTR-Nogo-A-mut, and empty pmiRGLO. Bar graphs show mean ± SEM of the firefly/Renilla emission ratio normalized to the corresponding pmiRGLO without mimic co-transfection (pmiRs control) in Neuro-2a cells (two-tailed paired *t*-test, T_4_ = 4.789, *p* = 0.0087, n = 5 independent experiments). ** indicates a *p*-value < 0.01. (**B**) Representative immunoblot of Nogo-A and the load-control α-tubulin protein expression in Neuro-2a cells and C6 cells transfected with either miR-182-5p or negative control mimics. Bar graphs show the mean ± SEM of three independent experiments; Nogo-A expression was normalized by the respective α-tubulin expression, taking negative control values as control levels. miR-182-5p mimic transfection showed statistical reduction in the Neuro-2a cell line (two-tailed paired *t*-test, T_2_ = 7.291; *p* = 0.0183) and the C6 cell line (two-tailed paired *t*-test, T_2_ = 4.755; *p* = 0.0415). * indicates a *p*-value < 0.05.

**Figure 4 pharmaceuticals-15-00529-f004:**
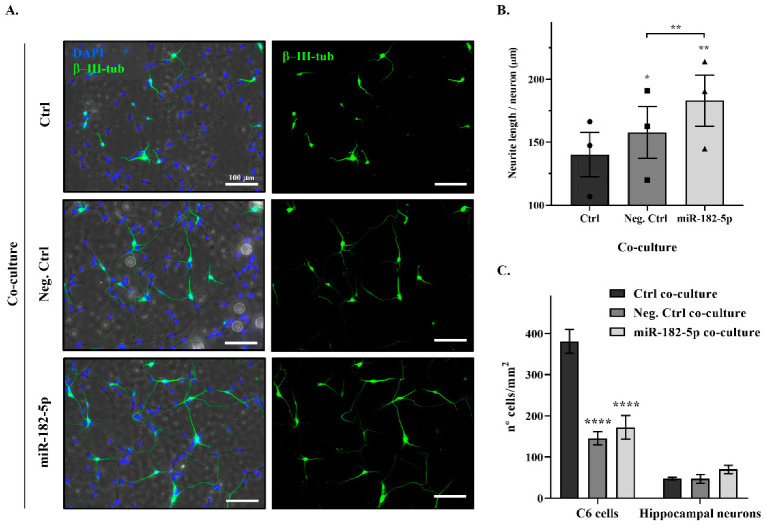
Nogo-A downregulation by miR-182-5p mimic promotes neurite outgrowth of primary hippocampal neurons. (**A**) Representative phase contrast and epifluorescence images of different co-cultures of C6 cells (non-transfected control cells; negative control mimic transfected cells; miR-182-5p mimic-transfected cells; imaged in phase contrast) and rat hippocampal neurons, labeled with the specific neuronal marker β-III tubulin (green) and DAPI (nuclei staining, blue). Bar scale = 100 μm. (**B**) Bar graph shows the mean ± SEM of three independent experiments of neurite length per neuron (µm) of mimic-transfected C6 cell co-cultures in comparison with the non-transfected C6 cell co-culture values (two-tailed paired *t*-test; negative control co-culture, T_2_ = 5.041, *p* = 0.0372; miR-182-5p co-culture, T_2_ = 15.46, *p* = 0.0042) and the comparison between both mimics-transfected C6 cell co-cultures (two-tailed paired *t*-test; T_2_ = 17.65, *p* = 0.0032). * indicates a *p*-value < 0.05, and ** indicates a *p*-value < 0.01. (**C**) Bar graph shows the mean ± SEM of three independent experiments of the evaluation of the number of C6 cells and hippocampal neurons per mm^2^ (density) in the different co-cultures. Mimic transfection showed statistical reduction in the number of C6 cells (Tukey’s multiple comparison post hoc test, *p* < 0.0001). **** indicates a *p*-value < 0.0001.

**Table 1 pharmaceuticals-15-00529-t001:** Primers used for subcloning of 3′UTR-Nogo-A-wt and 3′UTR-Nogo-A-mut fragments, as well as for DNA sequencing.

**Primers**	**Sequences (5′-3′)**
3′UTR-Nogo-A-wt	Forward: AACGAGCTCCATTCATCTTTAAAGGGGAC Reverse: ATATCTAGATTATGTCTATAT
3′UTR-Nogo-A-mut	Forward: GTTAGAGAATTCATATAAGTAAATATAG Reverse: CTTATATGAATTCTCTAACAGTAAATC
pmiRGLO sequencing	CAAGAAGGGCGGCAAGATCG

## Data Availability

Data is contained within the article.

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
