# Peer review of "miR-182-5p Regulates Nogo-A Expression and Promotes Neurite Outgrowth of Hippocampal Neurons In Vitro"

_pharmaceuticals, 2022, doi:10.3390/ph15050529_

Round 1

Reviewer 1 Report

The reviewed manuscript describes a novel therapeutic approach to diseases associated with axonal pathology. The authors succinctly describe a series of experiments that demonstrate the regulatory effect of miR-182-5p on the Nogo-A axonal regeneration inhibition promoting neurite outgrowth in murine neural cell lines, thus providing data to search for new treatments for patients with brain damage, Parkinson's, Alzheimer's and other diseases. The study is clearly described, concise and very interesting. The materials and method section is described clearly and sufficiently, allowing other researchers to use this methodology. I did not find any serious flaws in the study design, methodology or overall merit, however some minor corrections are needed to successfully publish this article in Pharmaceuticals.

Minor comments:

  1. Line 52: [11] is enough, please omit the text in parentheses.
  2. Lines 99-103: it seems like data repetition form Figure 1, is it necessary?
  3. In my opinion, all references mentioned in the Results section should be moved to the Discussion section and discussed there. The results should only be a presentation of the pure results obtained by the authors.
  4. Lines 132-134, 151-155, 201-204: this text belongs to M&M section.

Author Response

Reviewer 1

Minor comments:

  1. Line 52: [11] is enough, please omit the text in parentheses.
    • Thanks for the comment. We have modified the text as suggested. The sentence now reads as: “ …and models of multiple sclerosis [11].”
  1. Lines 99-103: it seems like data repetition form Figure 1, is it necessary?
    • To eliminate unnecessary repetition we have deleted the sentence describing the number of targets identified by each algorithm. The paragraph now reads as: “A bioinformatics-based prediction of the potential targets of miR-182-5p in mouse mRNAs was performed. Since the various available programs can yield rather different predictions, we combined miRmap, miRanda 3.3a, TargetScan 8.0 and miRWalk 3.0- programs to search for mouse miR-182-5p gene targets (Figure 1A). A total of common 191 genes are identified by the four prediction programs, including Nogo (Rtn4 gene).”
  1. In my opinion, all references mentioned in the Results section should be moved to the Discussion section and discussed there. The results should only be a presentation of the pure results obtained by the authors.
    • Following the suggestion by the reviewer, we have moved the references and involved sentences to the first paragraph of the discussion which now indicates that: “Amongst many factors, one of the major inhibitory signals of the CNS environment to regrowth is the myelin-associated Nogo pathway, which plays an important role in regeneration [27-29]. Oligodendrocytic Nogo-A binds to its neuronal receptor (NgR1) and co-receptors (LINGO-1 and p75NTR or TROY) producing the inhibition of neurite outgrowth of the neurons [8-10]. In the present study, we describe and validate for the first time Nogo-A post-transcriptional regulation by a miRNA in two murine neural cell lines, Neuro-2a and C6, which have been extensively used to study neuronal differentiation and axonal growth [30, 31]. We demonstrate that miR-182-5p downregulates Nogo-A expression promoting neurite outgrowth of primary hippocampal neurons in vitro.”
  2. Lines 132-134, 151-155, 201-204: this text belongs to M&M section.

We have proceeded according to the reviewer suggestion as follows:

    • lines 132-134: we have reduced and reorganized the paragraph to reduce the methodological information. The paragraph now reads as: “We chose Neuro-2a and C6 cell lines to study Nogo-A regulation by miR-182-5p mimic in vitro due to their Nogo-A protein expression levels (Figure 2A) and endogenous expression of miR-182-5p (Figure 2B). Conversely, SH-SY5Y cell line was discarded because Nogo-A protein is undetected in its extracts.”
    • lines 151-155: we have deleted the involved paragraph because it is already described in detail at the material and methods section.
    • lines 201-204: we have reorganized the paragraph and deleted some sentences. However, we have kept the initial description of the methodological approach to allow the reader to understand the results that are described below. The paragraph now indicates that: “To determine the biological effects of the downregulation of Nogo-A by miR-182-5p transfection on neurite outgrowth, we performed functional analyses based on the co-culture of rat primary hippocampal neurons with C6 cells transfected with either miR-182-5p or Neg. Ctrl mimics. Transfection of C6 cells with miR-182-5p mimic increases significantly neurite length of primary hippocampal neurons… “

Reviewer 2 Report

The manuscript "MiR-182-5p regulates Nogo-A expression and promotes neurite outgrowth of hippocampal neurons in vitro", written by Soto A, Nieto-Diaz M, Reigada D, Barreda-Manso MA, Munoz-Galdeano T and Maza RM, describes the effect of miR-182 on Nogo-A expression and consequently on the outgrowth of the neuronal neurites. The authors found, by bioinformatics analysis, that miR-182 could influence NogoA expression and analyzed several neural cell lines for NogoA and miR-182 expression. Furtheron, they analyzed effect of miR-182 mimics on 3'UTR Nogo A using dual luciferase system, and checked for Nogo A expression in transfected cells. At the end they analyzed the effect of miR182 transfected C6 cells on the neurite outgrowth of primary hippocampal cells in coculture.

The manuscript is well written, Introduction is informative, and Results are clearly presented with all necessary details. Also, Materials and methods and detailed. I would just comment functional experiments of neurite outgrowth. According to data there was significant outgrowth in neurons coculturing with negative control cells. To have more objective picture, Figure 4 C could present all three cocultures. If transient transfection is used, maybe cells were under stress, 24 h after, and release some molecules which influence neural cells. I would suggest to produce stable clones of C6 or other cells, overexpressing miR-182, and repeat the experiment with these cells.

Minor comments

line 66: ...has also been shown to promote...

lines 67, 86: genes should be written in italics

data in paragraph 210-213 seems to me repeated in next paragraph

line 406: data were

Author Response

Reviewer 2

Major Comment

  1. I would just comment functional experiments of neurite outgrowth. According to data there was significant outgrowth in neurons coculturing with negative control cells. To have more objective picture, Figure 4 C could present all three cocultures. If transient transfection is used, maybe cells were under stress, 24 h after, and release some molecules which influence neural cells. I would suggest to produce stable clones of C6 or other cells, overexpressing miR-182, and repeat the experiment with these cells.
    • We have tried to fulfil the suggestions commented by the reviewer. On the one hand, we have modified the figure, incorporating the data of the non-transfected C6 cells (Ctrl co-culture) to represent all conditions under analysis and modifying the results description accordingly. The figure (now Figure 4B) and the corresponding caption are now as follows:

-(See figure 4.B in the final version of the manuscript)

-Caption: “B) Bar graph shows the mean ± SEM of three independent experiments of neurite length per neuron (µm) of mimics-transfected C6 cell co-cultures in comparison with the non-transfected C6 cell co-culture values (two-tailed paired t-test; Neg. Ctrl co-culture, T2=5.041, p=0.0372; miR-182-5p co-culture, T2=15.46, p=0.0042) and the comparison between both mimics-transfected C6 cell co-cultures (two-tailed paired t-test; T2=17.65, p=0.0032). * indicates a p-value < 0.05 and ** a p-value < 0.01. C) Bar graph shows the mean ± SEM of three independent experiments of the evaluation of the number of C6 cells and hippocampal neurons per mm2 (density) in the different co-cultures. Mimics transfection showed statistical reduction in the number of C6 cells (Tukey’s multiple comparisons post-hoc test, p<0.0001). **** indicates a p-value < 0.0001.”

  • On the other hand, we agree with the reviewer that stress-induced paracrine regulation may be an alternative transfection-driven mechanism that can contribute to or even mask the effects of miR-182-5p. However, such paracrine effect is unlikely considering that the culture medium is changed 24 hours after C6 transfection and just before neuron seeding. In any case, the analysis of such a paracrine mechanism is beyond the scope of this article. We also agree that stable cells, particularly conditional ones, may help to understand the contribution of transfection. Indeed, the actual results clearly demonstrated the neurite growth promoting effects of transfecting miR-182-5p in C6 cells (by comparison with the effects of the negative control transfected cells) beyond any potential effects of the transient transfection itself. Anyway, we have acknowledged the potential role of paracrine regulation by miR-182-5p overexpressing cells and the potential of stable lines overexpressing this microRNA in the discussion as follows: “Although the role of miR-182-5p as a regulator of neurite outgrowth has been described in cortical and midbrain neurons through activation of the PTEN/AKT pathway [23, 58], our results could provide a broader implication with regards to the axonal regeneration. Our functional assays (Figure 4) showed that the downregulation of Nogo-A due to miR-182-5p overexpressed in neural cells eased the neurite outgrowth of primary hippocampal neurons. However, a better understanding of the miR-182-5p regulation on Nogo-A including exploring non-canonical mechanisms (vg. paracrine regulation) would be needed to establish its precise role following CNS injury. Analyses would greatly benefit from gain- and loss-of-function assays employing stable and conditional cell lines.”

Minor comments

  1. Line 66: ...has also been shown to promote…
    • We have modified the text as suggested by the reviewer. The text now says that: “...or miR-133b which has also been shown to promote neurite outgrowth in primary cortical neurons…”
  1. Lines 67, 86: genes should be written in italics
    • Thanks for the comment. We have modified the text as suggested by the reviewer. We have also reviewed the full text and corrected the format of several other genes as indicated.
  1. Data in paragraph 210-213 seems to me repeated in next paragraph.
    • Thanks for the observation. We have modified the paragraph to avoid unnecessary repetition. The text now reads as: “... Transfection of C6 cells with miR-182-5p mimic increases significantly neurite length of primary hippocampal neurons in ~26 µm per neuron (miR-182-5p co-culture) in comparison with the Neg. Ctrl transfected C6 cells (Neg. Ctrl co-culture) (183.0 ± 20.32 µm vs. 157.7 ± 20.59 µm Neg. Ctrl co-culture; two-tailed paired t-test, T2=17.65, p=0.0032) (Figure 4A and B). Moreover, transfection of C6 cells with both Neg. Ctrl and miR-182-5p mimics increases neurite length in comparison with the non-transfected C6 cells (Ctrl co-culture)…”
  1. Line 406: data were
    • We have modified the text as indicated. The text now says that: “Firefly emission data were normalized to Renilla load control levels and expressed as firefly/Renilla ratio.

Following the editor's advice, we have also included a conclusions section.

Round 2

Reviewer 2 Report

The authors responded to the comments. I think the manuscript can be published in the present form.

Minor comments:

line 242: omit comma

line 299: single gene